# Botulinum Toxin: An Update on Pharmacology and Newer Products in Development

**DOI:** 10.3390/toxins13010058

**Published:** 2021-01-14

**Authors:** Supriyo Choudhury, Mark R. Baker, Suparna Chatterjee, Hrishikesh Kumar

**Affiliations:** 1Department of Neurology, Institute of Neurosciences Kolkata, Kolkata 700017, India; choudhurydrsupriyo@gmail.com (S.C.); drsupchat@gmail.com (S.C.); 2Departments of Neurology and Clinical Neurophysiology, Royal Victoria Infirmary, Queen Victoria Rd, Newcastle upon Tyne NE1 4LP, UK; Mark.Baker@newcastle.ac.uk; 3Translational and Clinical Research Institute, Newcastle University, Framlington Place, Newcastle upon Tyne NE2 4HH, UK; 4Department of Pharmacology, Institute of Post Graduate Medical Education and Research, Kolkata 700020, India

**Keywords:** botulinum toxin, dystonia, recombinant botulinum toxin, acetylcholine, neuromuscular blockade

## Abstract

Since its introduction as a treatment for strabismus, botulinum toxin (BoNT) has had a phenomenal journey and is now recommended as first-line treatment for focal dystonia, despite short-term clinical benefits and the risks of adverse effects. To cater for the high demand across various medical specialties, at least six US Food and Drug Administration (FDA)-approved formulations of BoNT are currently available for diverse labelled indications. The toxo-pharmacological properties of these formulations are not uniform and thus should not be used interchangeably. Synthetic BoNTs and BoNTs from non-clostridial sources are not far from clinical use. Moreover, the study of mutations in naturally occurring toxins has led to modulation in the toxo-pharmacokinetic properties of BoNTs, including the duration and potency. We present an overview of the toxo-pharmacology of conventional and novel BoNT preparations, including those awaiting imminent translation from the laboratory to the clinic.

## 1. Introduction 

More than two centuries ago, Justinus Kerner, a young German physician, suggested the putative clinical utility of a toxin extracted from bad sausages (*botulus*, Latin for sausage), which had caused a number of fatal outbreaks in the kingdom of Württemberg in the late eighteenth and early nineteenth centuries [1]. However, the first documented therapeutic application of botulinum toxin was not until 1977, when Dr. Alan B. Scott injected a purified botulinum toxin (Oculinum©) into extra-ocular muscles to treat strabismus [2].

Botulinum toxin (BoNT) was first licensed for use in 1989 by the US FDA for the treatment of strabismus [3]. Since then, there has been a surge of research into botulinum toxins, which has led to the addition of newer formulations with an increasing range of indications. To date, BoNT has been widely used by neurologists and cosmetic practitioners [4]. With the rapid expansion in the number of therapeutic indications for BoNT, the first trade name—“oculinum”—was changed to “botox” within the first two years of its introduction to the market [1]. More recently, urologists and pain specialists are increasingly using BoNT for various novel indications [5,6]. Nevertheless, the application of BoNT to neurological disorders is probably the most common therapeutic indication.

A number of innovative BoNT formulations have been developed in the last few years. In addition to the naturally derived products, synthetic, genetically engineered products are also now available. There are some differences in the pharmacological profile of the formulations currently available, which merit review. Currently, there are more than six different formulations approved for clinical use, and more are in the pipeline. It is important that prescribers should be aware of the characteristics of these formulations to make an informed decision when prescribing. Additionally, such information is essential when the interchangeability of formulations is being considered.

This review will cover the salient toxo-pharmacological profiles of BoNTs, types of the conventional BoNT formulations, and the comparative toxo-pharmacology of newer BoNT (marketed, those in pipeline) with respect to older products.

## 2. Structure and Types

BoNTs are produced by the anaerobic, spore-forming bacteria *Clostridium botulinum, Clostridium*
*butyrricum, Clostridium barati*, and *Clostridium argentinensis,* and a few other related species [7]. Whilst all BoNTs are composed of two peptide chains linked through a disulphide linkage, there are significant differences in the amino acid sequence of the various peptide chains found in each subtype of BoNT. The molecular weight of the heavy chain is 100 k Dalton, and that of the light chain is 50 k Dalton [8]. The entire protein comprises three domains—two in the heavy chain and one in the light chain [9], and each domain performs a specific function at the molecular level. The C-terminal of the heavy chain is involved in binding the toxin to the receptor site, whereas the N-terminal is responsible for a function known as “translocation” (described subsequently) [10]. The light chain contains the catalytic unit. Apart from these two peptide chains, the toxin molecule is typically surrounded and stabilised by a set of naturally occurring proteins (≈750 k Dalton molecular weight) (Figure 1) [11].

Conventionally, seven types of BoNTs are described in the literature, from A to G [12], which are classified based on the serological typing of the toxins. In other words, the type is determined by the specific neutralising anti-sera. BoNT/A, BoNT/B, BoNT/E, and BoNT/F cause botulism in both humans and animals, whereas BoNT/C and BoNT/D cause disease only in domestic animals [13]. BoNT/G-producing organisms have been isolated from soil but never reported to the cause of botulism [14]. Subsequently, various subtypes were identified and labelled through alpha numeric suffix following the serotype of toxin [15]. These subtypes are based on the specific variation in amino acid sequence within a particular serotype of toxin. The serotypes and subtypes not only differ structurally, but significant differences are also apparent in their toxo-pharmacological properties [16].

## 3. Mechanism of Action of BoNT at the Neuromuscular Junction

The functional unit of skeletal muscle contraction comprises the motor end plate, which is the junction between the motor neuron and the muscle fibre. Acetylcholine (ACh) is released from the terminals of motor axons when action potentials, generated at the initial segment of the motor neuron within the central nervous system, arrive at the terminals. Then, the muscle fibres contract when ACh, which binds to and opens a specific ionotropic receptor (the nicotinic cholinergic receptor) on the muscle fibre, depolarises the post-synaptic membrane [17]. Botulinum toxin essentially blocks the release of ACh from the motor terminals, and hence, skeletal muscles fail to contract even though action potentials continue to reach the motor end plate [18].

There are well-defined stepwise activities of botulinum toxin on the neuro-muscular junction, as depicted in Figure 2.

Firstly, the receptor binding domain of the heavy chain binds to polysialogangliosides (PSGs) on the cell surface. Subsequently, the toxin is internalised through binding with another surface receptor, either synaptotagmin (Syt) or Glycosylated Sv2. BoNT/B1, BoNT/DC, and BoNT/G specifically binds with Syt, whereas BoNT/A1 and BoNT/E1 bind with Sv2 [19,20,21,22]. After internalisation, the toxin resides within synaptic vesicles. Then, the vesicles are acidified by the influx of the H ^+^ ion through vesicular proton pumps, thus activating ACh transporter proteins in the vesicle membrane, which import and concentrate cytosolic ACh within the vesicle. At this stage, in the absence of BoNT, the vesicles are ready to fuse with the presynaptic membrane and release ACh into the synaptic cleft. However, botulinum toxin interferes with the steps of release thereafter. First, the light chain is “translocated” to the cytoplasm from inside the vesicles, which is facilitated by the N terminal of the heavy chain (translocation domain). The light chain remains inactive whilst it remains bound to the rest of the toxin. After translocation, the light chain is released by the action of cleaving enzymes such as heat shock protein 90 (hsp90) and the thioredoxin reductase–thioredoxin system (TrxR-Trx). The free and active light chain now cleaves and deactivates various proteins such as VAMP, SNAP25, and syntaxin, which are essential for the release of ACh. These proteins (SNARE proteins) are essential for the fusion of vesicles with the presynaptic membrane and subsequent release of the toxins into the synaptic cleft. The cleaved target proteins are specific to the type of BoNT. BoNT/B, BoNT/D, BoNT/F, and BoNT/G cleave VAMP, BoNT/A and BoNT/E cleave SNAP-25, and BoNT/C cleaves both SNAP-25 and syntaxin (Stx). By inactivating these proteins, BoNT blocks the release of ACh, resulting in reversible chemical paralysis of the muscles. The duration of paralysis depends on the half-life of the light chain and turnover time of SNARE proteins [11,23].

The effects of BoNT are not exclusive to the cholinergic terminals of the neuromuscular junction, and a more general effect on neurotransmission at chemical synapses in both the peripheral and central nervous system is generally accepted. Thus, neurotransmitters affected by the actions of BoNT include molecules in small synaptic vesicles (e.g., acetylcholine and glutamate) and neuropeptides in large dense core vesicles (e.g., calcitonin gene-related peptide (CGRP), pituitary adenylate cyclase activating peptide 38 (PACAP 38), and Substance P). Large dense core vesicles also carry cargo including proteins and receptors (e.g., transient receptor potential cation channel subfamily V member 1 (TRPV1), transient receptor potential cation channel subfamily A member 1 (TRPA1), purinergic receptor P2X ligand-gated ion channel 3 (P2 × 3), etc.), whose insertion into the lipid bilayer of the synaptic membrane is critical to nociception [24].

## 4. Salient Prescribing Information of Conventional Botulinum Toxins

The first botulinum toxin approved for human therapeutic use was licenced in 1989 in the USA [25]. Currently, there are six US FDA-approved formulations available on the market. There are some subtle differences in the toxo-pharmacological properties of the formulations, which should preferably not be used interchangeably [26]. Therefore, knowledge of the toxo-pharmacological properties of individual formulations is imperative for prescribers. Table 1 lists the available toxin formulations along with their approved indications. The proprietary conventional BoNT products are *onaBoNT, aboBoNT*, *incoBoNT,* and *rimaBoNT.* The first three products are type A1 BoNT, while *rimaBoNT* is serotype B [27].

The BoNTs are administered by intramuscular injection into affected muscles or in other targeted tissues, such as directly into the salivary glands in adults with sialorrhea. The maximum dose used is much less than the lethal dose (3000 U of Botox in monkey) [28]. Table 2 summarises the manufacturing process and pharmaceutical preparation of each formulation.

Most of the formulations are lyophilised or vacuum dried, so they need to be reconstituted with normal saline except for *rimaBoNT*, which is available as a solution for injection [29]. The quantity of toxin also varies across formulations. Therefore, the unit of injection is also variable. For example, *onaBoNT* (botox) is available in 50/100/200 U vials, whereas *aboBoNT* (dysport) is available in 300/500 U vials. The dose of dysport is usually 2.5 to 3 times that of botox [23,30]. Naturally, the injector should check and confirm the label information before injection. Interestingly, the size of the toxin complex also varies across formulations. It was previously thought that the molecular size of the toxin complex determined its diffusivity in tissue. Botox has the highest complex size of 900 kDa, whereas dysport is around 500 kDa. However, experimental studies have shown that diffusivity is unrelated to the size of the toxin complex [31].

Very rarely, BoNT is found to be ineffective, although the reported incidence is less than 1% [32]. The production of neutralising host antibodies against the toxin could be one of the reasons for the lack of efficacy. It has been suggested that the excipient used in stabilising the toxin, human serum albumin (HSA), might induce the immunological destruction of the toxin [23]. Therefore, efforts have been made to reduce the amount of HSA in toxin formulations. Notably, *incoBoNT* has the lowest HSA concentration among the four US FDA-approved products [27], although the clinical benefit of reducing HSA is yet to be established.

It is quite clear that the formulations are not identical or equivalent. There have been head to head comparisons of formulations. Some formulations (e.g., *Dysport*) were found to have a longer duration of action but increased adverse effects outside the target site compared to *Botox*, which were possibly related to the higher quantities of neurotoxin in *Dysport*. Injection volume, toxin concentration, and dose may all play significant roles in the therapeutic and non-therapeutic effects of individual formulations [27]. The *incoBoNT (Xeomin)* has its own advantages over other formulations. For example, it does not need refrigeration, and negligible amounts of albumin (protein load of 0.44 ng/100 unit) are present in the formulation [33]. Thus, the theoretical risk of antibody production against the toxin is less than other formulations. Additionally, reconstituted *Xeomin* also does not show a reduction in potency over 52 weeks of treatment [34]. *Botox* has also shown reasonable stability after reconstitution. The stability of reconstituted toxins is important when single vials are shared between patients, which is an approach that reduces the “out of pocket” expenses for one patient. Notably, reconstitution is not needed for *Myobloc*/*Neurobloc* [23].

There are two other products that are widely used and approved in China and Korea. The brand names of the products are *Prosigne* and *Meditoxin*, respectively. The main excipients of Prosigne (per vial) are gelatine 5 mg, dextran 25 mg, and sucrose 25 mg with a minimum protein load of 4–5 ng/100 units. The potency of *Prosigne* is close to that of *Botox* (1:1/1:1.5). *Meditoxin* has almost an identical structure to *Xeomin* with a low protein load. Moreover, it does not require reconstitution [23] and can be kept at room temperature. In 2013, Allergan purchased the license for *Meditoxin* for future distribution in the USA, and various Phase III US FDA trials are underway [29].

## 5. Newer Botulinum Toxin Currently in Development Stage

A number of newer formulations have been approved recently for market or are in the late stages of development. For example, Revance announced that the US FDA had accepted its Biologics License Application (BLA) for DAXI for the treatment of glabellar (frown) lines on 6th February 2020 [35]. A summary of the newer formulations is described in Table 3.

Among these, BoNT/E has rapid onset but a short duration of action (2–4 weeks) [36]. This unique property might be useful for pain management in conditions such as osteoarthritis. *PraBoNT* was initially given a brand name of *JEUVEAU (Ju-vo)* inspired by the word “nouveau” (nu-vo)—from the French for “new”. Subsequently, the manufacturer changed the brand name to *Neuronox* [37], which has also been approved for cosmetic use (US FDA approved) since 2019 [38]. Table 4 summarises recent clinical trials of the newer BoNTs. In late-stage clinical trials, *daxiBoNT* and *letiBoNT* were found to be effective for various non-cosmetic indications, including movement disorders [36].

## 6. Recombinant Botulinum Toxin and Application of In Silico Drug Development

In the era of next-generation sequencing and recombinant technology, the identification of novel variants of BoNTs and the production of genetically engineered BoNTs have reached new heights. It is now possible to data mine the genetic sequence from existing databases and find novel proteins that align with the known sequence of BoNT variants. Genetically engineered BoNTs are essentially produced by smuggling the coding section of DNA (open reading frame, ORF) of BoNTs into *E. coli* and other microbes [52]. Microbes with the incorporated BoNT ORF subsequently produce BoNT along with their own proteins. The potential applications of genetically engineered BoNTs, using highly characterised toxicological reference material, are manifold. These developments are important not only from the perspective of quality assurance but because of the potential to produce designer drugs; by altering genetic sequences, it is theoretically possible to change the toxo-pharmacological properties of the toxins, and thus, non-toxic BoNTs could be engineered to produce vaccines or toxoids [36].

One of the initial variants of BoNT identified through in silico data mining of gene sequences was BoNT/H. This was identified from a toxin reported to cause infant botulism in 2014 by the *C. botulinum* strain IBCA10-7060 [53]. Initially, the authors presumed that it was a novel variant, a bivalent strain of B2 toxin, which was denoted as serotype H. In early experiments, it was found that the toxin could be weakly neutralised by antibodies against currently known serotypes. However, subsequent testing demonstrated its elimination by serotype A antitoxin. Sequence analysis of the translated BoNT/H ORF (open reading frame) indicated ≈80% homology of the LC fragment with the BoNT/F5 LC and 64% homology of the HN segment with BoNT/F1, and the receptor binding domain (RBD) shared 84% homology with BoNT/A1 [54]. Hence, it was concluded that it was not a novel serotype but a chimeric protein of BoNT/F and BoNT/A that can cleave VAMP-2 between L(Leucine) 54 and E (Glutamic Acid) 55. In various assays, it was reported that the potency of this chimeric toxin is 5 to 20 fold lower than the activity of BoNT/A [55,56].

BoNT/A and BoNT/B were first identified in 1919 by Georgina Burke. The last of the seven serotypes, BoNT/G, was discovered in 1969 [57]. Around half a century after this discovery, Zhang et al. discovered a novel eighth serotype of BoNT, which is isolated from the *C. botulinum* strain 111 through a bioinformatics approach [57]. To validate its activity, a small amount of full-length BoNT/X was assembled by linking two non-toxic fragments using a transpeptidase (sortase) [57], and this was shown to cleave VAMP2 and VAMP4 in cultured neurons and cause flaccid paralysis in mice. Moreover, the cleavage of VAMP-2 occurred at a novel site, between R (Arginine) 66 and A (Alanine) 67 [57]. It is non-reactive to any known anti-toxin from serotype A–G, and it has much lower potency in vivo.

## 7. Non-Clostridial Botulinum Toxins

By examining bioinformatics databases in silico, a number of proteins similar in structure and properties to the botulinum toxins have been identified. Interestingly, some of the proteins identified were isolated from non-clostridial microbial species [58]. A summary of their characteristics is presented in Table 5.

## 8. Chimeric Botulinum Toxins

Different approaches have been used to change the properties of BoNTs according to clinical need. One such approach is to engineer chimeric proteins from two different BoNT serotypes or subtypes, as demonstrated by one study published in 2008, where the production of two recombinant chimeric proteins using an *E. coli* codon, BoNT/EA (A LC-HN, E RBD) and BoNT/AE (E LC-HN, E RBD), was reported [54]. Both chimeric proteins retained the basic BoNT functions, but the time to paralysis was different, and the potency was lower compared with either of the parental toxins. The recombinant toxins were found to cleave SNAP-25, and recovery took up to 37 days with the AE chimera [54]. BoNT/EA was also found to block the release of the capsaicin-evoked pro-inflammatory calcitonin gene-related peptide (CGRP) [54] and thus a potential treatment for migraine type headache [62].

Another pair of chimeric BoNTs were subsequently developed, namely BoNT/AB (A LC-HN/B RBD) and BoNT/BA (B LC-HN/A RBD) [63]. Both are equivalent in their ability to induce paralysis. The BoNT/BA chimera is twice as toxic as BoNT/A and 20-fold more toxic than the BoNT/AB. The BoNT/AB chimera was found to cleave significantly more SNAP-25 than the parental BoNT/A and resulted in longer lasting paralysis. The BoNT/AB chimera is 8.4 times more potent than recombinant BoNT/A [63].

Chimeric proteins are often composed of fused BoNT subtypes—for example, A1LC/A3HC and A3LC/A1HC [64]. These fused protein subtype toxins have provided insights into the roles of the LC and HC in terms of both the potency and duration of the toxic effects. The duration of paralysis appears to be influenced by the LC, whereas LC and HC in combination appear to determine potency.

## 9. BoNTs with Modified Target Specificity

The activity of BoNTs can also be modified by altering the BoNT amino acid sequence, using recombinant DNA technology. The modified BoNT might target other receptor proteins, according to clinical requirements. The feasibility of this approach is illustrated by observations of structural and functional changes resulting from naturally occurring point mutations in the BoNTs. In 2011, a recombinant BoNT/C1 with LC mutations was identified, which was unable to cleave SNAP-25 [65]. In a subsequent report, two mutants were found, which cleaved syntaxin 1A/1B with ≈10-fold less activity than the wild-type BoNT/C [66]. Tao et al. reported the production of an engineered BoNT/B designed to enhance binding to human synaptotagmin 2 (h-Syt II) in an effort to try and increase the therapeutic index of the toxin [67]. Mutation studies also create insights into the molecular mechanisms of the toxin. Cumulative data suggest that while SNAP-25 cleavage is required for the complete loss of neuromuscular transmission, syntaxin cleavage may contribute to recovery of the function and duration of paralysis. A synthetic *E. coli* synthesised BoNT/B containing the *E1191M/S1199Y* mutations, denoted as BoNT/BMY, was found to increase VAMP-2 cleavage [68], whereas an engineered, recombinant BoNT/B LC with a LC/T S’ pocket residue, *S201P* mutation was reported to enhance VAMP-2 cleavage tenfold. Moreover, BoNT/B1 *(S201P)* has arguably higher catalytic activity on VAMP 1 and VAMP 2 [68].

## 10. Conclusions

In this review, we have described the structure and molecular function of BoNTs. We started by describing the characteristics of conventional BoNTs before listing the newer BoNT formulations, including those that are in the later stages of development. The synthetic BoNTs are not yet used in the clinic but have huge potential in the treatment of various conditions. Clinical indications often demand specific characteristics of BoNT formulations; thus, genetically designed BoNTs are being increasingly considered. However, the prescriber needs to take into consideration the cost-effectiveness of the new formulations as well. Taken together, the new BoNTs hold promise, but evidence of the efficacy, safety, and cost effectiveness of such formulations from good quality clinical studies will guide future prescribing.

## Figures and Tables

**Figure 1 toxins-13-00058-f001:**
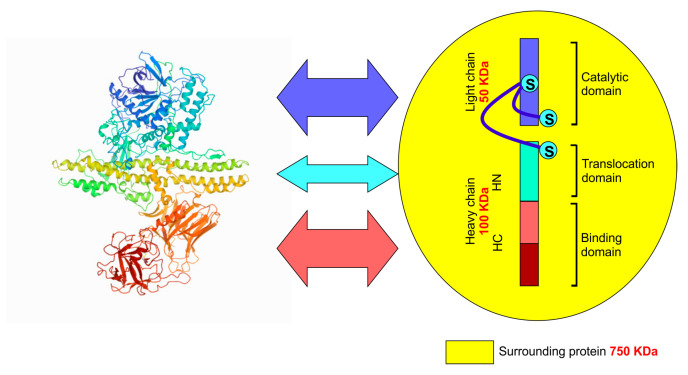
Schematic diagram and crystal structure of botulinum toxin type A. X-ray crystallography (PDB ID: 3BTA) shows the molecular organisation of botulinum toxin type A. The schematic representation shows that botulinum toxin type A has two peptide chains connected by a disulphide bridge. The heavy chain has two domains named after their specific activity (binding and translocation). The light chain is responsible for catalytic breakdown of the target protein. KDa = Kilo Dalton: S–S = disulphide bridge, HN = N terminal of heavy chain, HC = C terminal of heavy chain.

**Figure 2 toxins-13-00058-f002:**
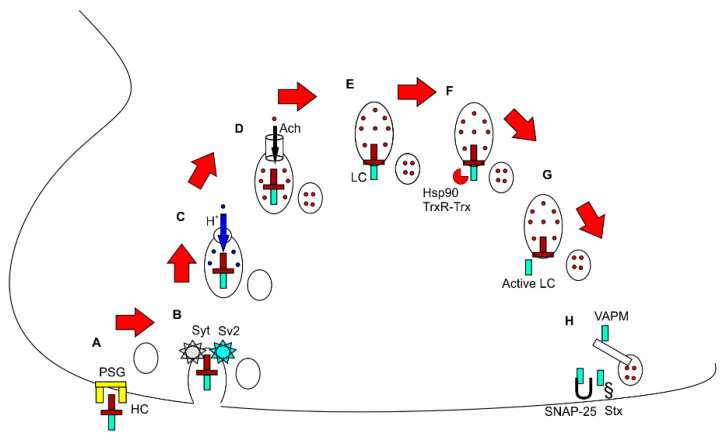
Molecular mechanism of botulinum toxin. (**A**–**H**) depict representative sequences of events within a synaptic terminal at the neuromuscular junction. (**A**) The heavy chain of botulinum toxin binds with the surface receptor; (**B**) The internalisation of the botulinum toxin is possible through its interaction with Sv2 or Syt; (**C**) Protons enter the synaptic vesicle through an active transporter; (**D**) The low pH inside the vesicle helps import Ach from the cytoplasm; (**E**) The translocation domain of botulinum toxin helps in the extrusion of botulinum toxin from the vesicle; (**F**) The catalytic enzymes act on the botulinum toxin; (**G**) The light chain is freed from the rest of the toxin; (**H**) The free and active light chain inactivates the target SNAP receptor (SNARE) proteins (SNAP25, Stx, VAMP). PSG = Polysialoganglioside. HC = Heavy chain, Syt = synaptotagmin, Sv2 = Synaptic vesicle protein 2, Ach = Acetylcholine, LC = light chain, Hsp90 = Heat shock protein 90, TrxR-Trx = Thioredoxin reductase–thioredoxin system, SNAP = soluble NSF attachment protein, NSF = N-ethylmaleimide sensitive fusion protein, SNAP 25 = Synaptosomal-Associated Protein, 25kDa, Stx = Syntaxin, VAMP = Vesicle-associated membrane protein.

**Table 1 toxins-13-00058-t001:** Conventional botulinum toxin formulations and its approved indications.

Trade Name	Proprietary Name	Manufacturer	US FDA Approved	US FDA Approved Indication	Year of Approval
Botox	OnabotulinumtoxinA	Allergan inc.	Yes	Blepharospasm	1989
				Hemifacial spasm	1989
				Strabismus	1989
				Cervical dystonia	2000
				Migraine	2010
				Upper limb spasticity	2010
				Lower limb spasticity (adult)	2014
				Bladder (NDO)	2011
				Bladder (OB)	2013
				Forehead wrinkles	2018
Xeomin	IncobotulinumtoxinA	Merz Pharmaceuticals	Yes	Cervical dystonia	2010
				Blepharospasm	2010
				Frown lines	2011
				Upper limb spasticity	2015
				Sialorrhea in adults	2018
Dysport	AbobotulinumtoxinA	Ipsen Pharmaceuticals	Yes	Cervical dystonia	2009
				Frown lines and wrinkles	2009
				Upper limb spasticity (adults)	2015
				Lower limb spasticity (children)	2016
				Lower limb spasticity (adult)	2017
Myobloc/Neurobloc	RimabotulinumtoxinB	US—WorldMed—Solstice	Yes	Cervical dystonia	2009
Proscine/ Prosigne	Type A	Lanzhou Institute, China	No		
Meditoxin/inotox	Type A	Meditox, South Korea	No		

**Table 2 toxins-13-00058-t002:** Molecular characteristics of conventional botulinum toxin preparations.

Proprietary Name	Serotype	Strain	Complex Size	Excipient	Stabilisation and Solubilisation	Unit/Vial	Neurotoxin Protein (ng/vial)
Botox (onabotulinumtoxinA)	A	Hall	900 kD	HSA (500 µg) Sodium chloride	Vacuum drying and normal saline	50, 100, 200	5
Xeomin (IncobotulinumtoxinA	A	Hall	150 kD	HSA (1 µg) Sucrose	Lyophilisation and normal saline	100, 200	0.6
Dysport (AbobotulinumtoxinA)	A	Hall	500 kD	HSA (125 µg) Lactose	Lyophilisation and normal saline	300, 500	4.35
Myobloc/Neurobloc (RimabotulinumtoxinB)	B	Bean	700 kD	HSA (500 µg/mL) Sodium succinate Sodium chloride solution	Solution	2000, 5000, 10,000	~25, 50, 100

**Table 3 toxins-13-00058-t003:** Newer botulinum toxin formulations either approved or under late phase of development.

Proprietary Name	Manufacturer	Trade Names (or Alternative names)	US FDA Approved	Advantages	Disadvantages
PrabotulinumtoxinA	Evolus, Inc. (USA)	Neuronox, Nabota	Yes, 2019; Glabellar Lines	Equivalent to botox Lower cost	
DaxibotulinumtoxinA	Revence Therapeutics (USA)	RT002	Yes, 2020; Glabellar Lines	No HSA Long duration (24 weeks)	
LetibotulinumtoxinA	Hugel Pharma (Korea)	Botulax	No		Lower potency than Xeomin
BotulinumtoxinE	BoNTi. Inc. (USA)	EB-001	No	Onset of action—24 h	Duration—2–4 weeks
Liquid Toxins	1. Medytox (Korea) 2. Galderma (Switzerland) 3. Allergan (USA)	Innotox	No	Lower risk of error in preparation	Costly

**Table 4 toxins-13-00058-t004:** Summary of recent literature and clinical trial reports on the newer botulinum toxins.

Serial Number	Author	Investigational Product	Study Design	Indication	Results
PrabotulinumtoxinA					
1	Beer KR et al. 2019 [39]	PrabotulinumtoxinA	Results from two identical phase III studies	Glabellar lines	Single dose of 20-U prabotulinumtoxinA was safe and effective for the treatment of glabellar lines.
2	Suh Y, 2019 [40]	PrabotulinumtoxinA with two different dosages	Multicenter, randomised, open-label comparative study	Gastrocnemius muscle hypertrophy	BTX at both dosages can be safely and effectively applied for calf muscle contouring without disturbing gait during walking or running.
3	Rzany BJ, 2020 [41]	Comparing PrabotulinumtoxinA and OnabotulinumtoxinA	Randomised, double-blind, placebo-controlled, single-dose, phase III, non-inferiority study	Moderate to Severe Glabellar Lines	A single treatment of 20 U prabotulinumtoxinA was safe and effective and noninferior to 20 U onabotulinumtoxinA for the treatment of moderate to severe glabellar lines.
4	Song S, 2018 [42]	Novel botulinum toxin type A (Nabota)	Single-arm, prospective, phase 4 clinical study	Glabellar frown lines	Onset of action was observed in the majority of subjects by 2 days after administration of Nabota. In addition, Nabota was found to be safe and effective for the treatment of glabellar frown lines.
Daxibotulinumtoxin					
1	Garcia-Murray E, 2015 [43]	RT002 (Daxibotulinum toxin)	Phase 1/2, open-label, sequential dose-escalation study	Glabellar lines	RT002 is a safe and effective BoNTA product with an extended duration of action.
2	Comella C, 2017 [44]	Daxibotulinumtoxin	Phase 2, open-label, dose-escalating study	Isolated cervical dystonia	DaxibotulinumtoxinA for injection up to 300 U in CD patients appears to be well tolerated.
3	Jankovic J, 2018 [45]	DaxibotulinumtoxinA	Phase 2, open-label (Level II), dose-Escalation Study	Isolated cervical Dystonia	The study shows that daxibotulinumtoxinA for injection (RT002) appears to be generally safe and well tolerated, and it may provide a long-lasting reduction in CD symptoms.
4	Truong D, 2018 [46]	DaxibotulinumtoxinA	Phase 2, dose-escalation study	Cervical dystonia	DaxibotulinumtoxinA appears to be generally safe and well tolerated, and it may provide a long-lasting reduction in CD symptoms
Letibotulinumtoxin					
1	Do KH, 2017 [47]	LetibotulinumtoxinA (BOTULAX^®^)	Randomised, double blind, multi-center, phase III clinical trial	Post stroke upper limb spasticity	The efficacy and safety of Botulax were comparable with those of Botox in the treatment of post-stoke upper limb spasticity.
2	Chang HJ, 2017 [48]	Letibotulinum toxin	Randomised controlled trial	Dynamic equinus foot deformity in children with cerebral palsy	Letibotulinum toxin A is as effective and safe as that of onabotulinum toxin A for the treatment of dynamic equinus foot deformity in children with spastic CP.
3	Kim JH, 2020 [49]	Letibotulinum toxin	Randomised controlled trial	Essential blepharospasm	Based on the study results, BOTULAX^®^ is considered to be an effective and safe treatment for essential blepharospasm.
4	Lee W, 2020 [50]	LetibotulinumtoxinA (BOTULAX^®^)	Retrospective study	Deviated nose and alar asymmetry	Botulinum toxin effectively restricted the paranasal muscles without any significant adverse events. We recommend injecting botulinum toxin after corrective rhinoplasty to prevent the recurrence of deviation by facial mimetic muscles.
Botulinum toxin E					
1	Yoelin SG, 2018 [51]	EB-001 (Botulinum toxin E)	Phase 2, randomised, placebo-controlled, ascending-dose study	Glabellar frown lines	In this clinical study of glabellar frown lines, EB-001 showed favorable safety, tolerability, and dose-dependent efficacy, with an 80% response rate at the highest dose. The maximum clinical effect of EB-001 was seen within 24 h and lasted between 14 and 30 days, which supports its development for aesthetic and therapeutic applications where fast onset and short duration of effect are desirable.

**Table 5 toxins-13-00058-t005:** Characteristics of non-clostridial botulinum toxins.

Year	Name	Authors	Organism	Genome	Recombinant Form	Mechanism of Action	Antisera
2015	BoNT/Wo	Mansfield, M.J. et al. [59]	*Weissella oryzae*, isolated from fermented Japanese rice	*SG25* genome	*E. coli* codon optimised ORFs encoding the LC and RBD were expressed and purified	Cleave recombinant rat VAMP-2 at the W89-W90 peptide bond	Weak cross -reaction with the anti-BoNT/C and the antiBoNT/D antisera
2018	eBoNT/J	Brundt et al. [60]	*Enterococcus* sp.	Novel BoNT gene cluster-*3G1_DIV0629*, with *ntnh* gene and *orfX* arrangement		Cleaves VAMP-2 between A67 and D68	
2018	BoNT/En	Zhang et al. [61]	*Enterococcus faecium* strain *IDI0629*, isolated from cow feces		A recombinant BoNT/En toxin was produced in limited amounts	Cleaves VAMP-2 between A67 and D68 SNAP-25 cleavage products indicated the cleavage occurs between K69 and D70	There was no observed cross-reactivity

## Data Availability

Not applicable.

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
