# Peer review of "Botulinum Toxin: An Update on Pharmacology and Newer Products in Development"

_toxins, 2021, doi:10.3390/toxins13010058_

Round 1
Reviewer 1 Report
The submitted manuscript comprehensively reviews the well-studied subtypes and clinically relevant formulations of the powerful neurotoxin botulinum toxin and advances new insights on novel modified forms of the toxin. The document is well written and structured, first focusing in extensively described and generally introductory topics such as toxin’s structure, types, mechanisms of action, commercially available formulations and their clinical indications followed by a summary of the most recent advances on toxin-related products, which is an innovative and very interesting subject.
Nevertheless, I recommend some corrections/ improvements that I am listing below:
- The 3rd topic addressed in the manuscript was entitled “Mechanism of Action of BoNT in Neurological Disorders” but authors exclusively described toxin’s activity in the neuromuscular junction. Neurological disorders not only comprise motor damages, but also potential changes on sensory, sympathetic and parasympathetic innervation. The title of the section should be corrected.
- Figure 2 is confusing and quite repetitive. A figure including the schematic and succeeding events of BoNT molecular mechanisms would be more informative and visually comprehensive. (Example image in Arnon S.S. et al., doi: 10.1001/jama.285.8.1059)
- In lines 110-111 the sentence might lead to misunderstanding. Better to write “After translocation, the light chain is released by the action of cleaving enzymes such as hsp90…”
- The effects of BoNT is not exclusive to cholinergic terminals and a wide effect on vesicles-mediated neurotransmission is currently accepted. Even if this is not the focus of the manuscript, authors should include a short paragraph on the topic as it also underlies the need for the development of new modified toxin products.
- Table 1 should be modified so that the years of toxin approval are in chronological order.
- In line 123 authors state that there are five US FDA approved formulations of the toxin and in Table 1 only four are listed.
- In line 132 authors wrote “The BoNTs are administered by intramuscular injection into affected muscles”. However, in some conditions the toxin might also be administered in other targeted tissues, such as directly into the salivary glands in adults with sialorrhea.
- In the conclusion, line 285 write “are not yet used in clinics”
Author Response
The submitted manuscript comprehensively reviews the well-studied subtypes and clinically relevant formulations of the powerful neurotoxin botulinum toxin and advances new insights on novel modified forms of the toxin. The document is well written and structured, first focusing in extensively described and generally introductory topics such as toxin’s structure, types, mechanisms of action, commercially available formulations and their clinical indications followed by a summary of the most recent advances on toxin-related products, which is an innovative and very interesting subject.
Thank you for your expert review. We have addressed the suggestions made by you in the manuscript.
Nevertheless, I recommend some corrections/ improvements that I am listing below:
- The 3rd topic addressed in the manuscript was entitled “Mechanism of Action of BoNT in Neurological Disorders” but authors exclusively described toxin’s activity in the neuromuscular junction. Neurological disorders not only comprise motor damages, but also potential changes on sensory, sympathetic and parasympathetic innervation. The title of the section should be corrected.
Changed in the manuscript
- Figure 2 is confusing and quite repetitive. A figure including the schematic and succeeding events of BoNT molecular mechanisms would be more informative and visually comprehensive. (Example image in Arnon S.S. et al., doi: 10.1001/jama.285.8.1059)
Changed as advised
- In lines 110-111 the sentence might lead to misunderstanding. Better to write “After translocation, the light chain is released by the action of cleaving enzymes such as hsp90…”
Changed in the manuscript
- The effects of BoNT is not exclusive to cholinergic terminals and a wide effect on vesicles-mediated neurotransmission is currently accepted. Even if this is not the focus of the manuscript, authors should include a short paragraph on the topic as it also underlies the need for the development of new modified toxin products.
The following paragraph is incorporated as advised.
“Vesicular contents include small molecules in small synaptic vesicles (eg, acetylcholine and glutamate), or neuropeptides in large dense core vesicles (eg, calcitonin gene-related peptide [CGRP], pituitary adenylate cyclase activating peptide 38 [PACAP 38], and Substance
P). Large dense core vesicle cargo include proteins and receptors (eg, transient receptor potential cation channel subfamily V member 1 [TRPV1], transient receptor potential
cation channel subfamily A member 1 [TRPA1], purinergic receptor P2X ligand-gated ion channel 3 [P2X3], etc.) whose insertion into the lipid bilayer of the synaptic membrane is critical for proper pain signaling (Burstein R et al. 2020).”
- Table 1 should be modified so that the years of toxin approval are in chronological order.
Thank you for pointing out. Now the table is in chronological order classified for individual toxin formulations.
- In line 123 authors state that there are five US FDA approved formulations of the toxin and in Table 1 only four are listed.
Thank you for identifying the discrepancy. Prabotulinum toxin and daxibotulinum toxin are two newer botulinum toxin formulations recently approved by USFDA. So, at present there are six US-FDA approved formulations.
We have adjusted the text accordingly.
- In line 132 authors wrote “The BoNTs are administered by intramuscular injection into affected muscles”. However, in some conditions the toxin might also be administered in other targeted tissues, such as directly into the salivary glands in adults with sialorrhea.
Added to the manuscript.
- In the conclusion, line 285 write “are not yet used in clinics”
Added to the manuscript.
Reviewer 2 Report
This article summarises history of toxin, the toxin available for clinical use nicely and give some good update on the immediate future in this field. This gives flavour of available toxin outside Europe and Americas too. This will be interesting to the readers to get an update in one article together. This will be nice addition to the existing literatures.Author Response
This article summarises history of toxin, the toxin available for clinical use nicely and give some good update on the immediate future in this field. This gives flavour of available toxin outside Europe and Americas too. This will be interesting to the readers to get an update in one article together. This will be nice addition to the existing literatures.
Thank you for carefully reviewing the manuscript and your encouraging words.